# A Unified Approach to Nested and Non-Nested Slots for Spoken Language Understanding

Xue Wan [1], Wensheng Zhang [1,2,*], Mengxing Huang [1,*], Siling Feng [1] and Yuanyuan Wu [1]

[1]  School of Information and Communication Engineering, Hainan University, Haikou 570100, China
[2]  Institute of Automation, Chinese Academy of Sciences, Beijing 100190, China
*   Correspondence: zhangwenshengia@hotmail.com (W.Z.); huangmx09@hainanu.edu.cn (M.H.)

**Abstract:** As chatbots become more popular, multi-intent spoken language understanding (SLU) has received unprecedented attention. Multi-intent SLU, which primarily comprises the two subtasks of multiple intent detection (ID) and slot filling (SF), has the potential for widespread implementation. The two primary issues with the current approaches are as follows: (1) They cannot solve the problem of slot nesting; (2) The performance and inference rate of the model are not high enough. To address these issues, we suggest a multi-intent joint model based on global pointers to handle nested and non-nested slots. Firstly, we constructed a multi-dimensional type-slot label interaction network (MTLN) for subsequent intent decoding to enhance the implicit correlation between intents and slots, which allows for more adequate information about each other. Secondly, the global pointer network (GP) was introduced, which not only deals with nested and non-nested slots and slot incoherence but also has a faster inference rate and better performance than the baseline model. On two multi-intent datasets, the proposed model achieves state-of-the-art results on MixATIS with 1.6% improvement of intent Acc, 0.1% improvement of slot F1 values, 3.1% improvement of sentence Acc values, and 1.2%, 1.1% and 4.5% performance improvements on MixSNIPS, respectively. Meanwhile, the inference rate is also improved.

**Keywords:** multiple intent detection; slot filling; MTLN; nested and non-nested; GP

## 1. Introduction

Intent detection (ID) and slot filling (SF) are two important parts of SLU [1,2], which are designed to recognize intents and capture semantic features for task-oriented dialogue systems such as Bixby, Siri, and Jovi. ID is a classification task, the purpose of which is to identify the intention of a user's dialogue. SF is the process of converting user's intentions into clear instructions to complement the information, which can be regarded as a sequence labeling task. We abandoned the BIO labeling form and directly used the slot type while annotating each slot at the first and last index positions of the text (e.g., *B-entity_name* → *entity_name* and *I-entity_name* → *entity_name*). As shown in Figure 1, the sentence *"add the keep your receipt ep to my digster reggae playlist and then play some dj qbert"* related to playing music and adding playlists is given, which contains multiple slots (*entity_name, playlist_owner, playlist, artist*) and two intents: *PlayMusic* and *AddToPlaylist*.

Early work focused on ID and SF modeling separately. As researchers constantly explored and analyzed, it was found that the independent modeling approach did not consider the correlation between the two subtasks, which results in a lack of semantics. Recently, researchers have gradually discovered that these two subtasks are closely related, and a slew of joint models [3–12] have proposed combining single ID and SF in the multi-tasking framework to capitalize on the information complementarity between intent and slot modules and solve a lack of semantics. Zhang and Wang [3] proposed a bidirectional gated recurrent unit (BiGRU) joint model based on conditional random field (CRF). In this model, BiGRU extracts the semantic features of the corpus, CRF decodes the slot information, and a maximum pooling layer acquires the global features of

the sentence for intent classification. With the wide application of attention mechanisms, a self-attention-based bidirectional recurrent neural network (BiRNN) joint model that captures the contextual information from around the current token was proposed by Liu and Lane [4]. Nevertheless, it simply combines the loss function of ID and SF without considering the information interaction between the two modules. For this consideration, Goo et al. [5] presented a slot-gating mechanism, which uses decoding intent to accomplish the SF task and achieves good performance. Wang et al. [6] suggested a BiRNN semantic frame parsing model, which uses BiRNN to decode intent and slot tasks, respectively, and share the hidden state information of each time step between two decoders. Niu et al. [7] continued to explore the information interaction between the two modules and proposed a bidirectional correlation model. Qin et al. [8] presented a model of stack propagation framework. In this model, the intent is decoded first, then the decoded intent information combines with the output value of the encoder for SF, and the slot is decoded finally. With the pre-training model performing well in all kinds of NLP tasks, Chen et al. [9] presented a BERT-based joint model that directly uses the classifier (CLS) to decode the intent and uses the token sequence to decode the slot and performs better. A typed abstraction method and a type iteration mechanism were introduced by Pang et al. [10] to achieve bidirectional encoding and reduce the interference of noisy information. In order to improve the multi-task model, He et al. [11] integrated an external knowledge base and constructed a loss-weight self-learning strategy. In consideration of the speed of model inference, Wu et al. [12] proposed a non-regressive joint model that improves the accuracy of model prediction by a two-pass mechanism. Meanwhile, the model's inference speed has been greatly improved.

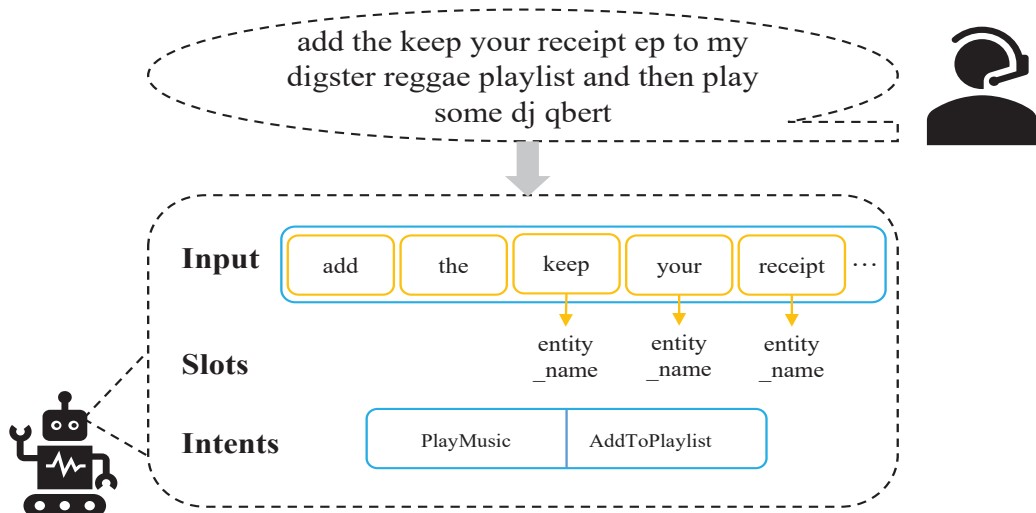

**Figure 1.** Illustration of a previous example of a joint modeling extracting multi-intent and slots.

Some research [13–21] has jointly modeled ID and SF on other datasets as well, and obtained good results. For example, Sun et al. [17] developed an iterative gating mechanism based on the interplay between intent and slot modules. Zhang et al. [19] proposed a joint and domain adaptive method based on Roberta and LSTM, which decodes the intent and slot information by calculating the distribution of attention between each token and the intent vector and between each token and the slot vector.

In a real-world dialogue system, the user's utterance usually contains multiple intents and corresponding semantic information, but the previous model is not suitable for multi-intent tasks. To solve this problem, Gangadharaiah and Narayanaswamy [22] made the first attempt to combine multi-intent and slot filling for modeling, and the model achieved good results. A self-distillation approach to complement the information of the intent module and the slot module was proposed in [23]. Qin et al. [24,25] constructed a graph interaction network to guide slot filling by GAT [26], which achieved better and faster results.

Although the existing multi-intent SLU joint models have made good progress, we find that they still have some problems:

(1) The above model either does not have high enough performance or the inference rate is not fast enough;

(2) The above model did not adequately consider the implied correlation between intentions and slots by filtering the information;

(3) At this stage, all the above models model slot filling as a non-nested task, so none of them can solve the slot nesting problem.

We provide a novel framework called MTLN-GP for combining multiple intent detection and SF to address the mentioned problems. Specifically, inspired by Jianlin Su [27], we constructed a multi-dimensional type-slot label interaction network (MTLN), which, with intention decoding, considers the implicit correlation between intents and slots to facilitate information interaction. At the same time, we used the global pointer network (GP) to solve the problem of nested slots and inconsistent slots. The proposed model delivers a state-of-the-art (SOTA) performance on a variety of indicators, according to experimental results on two public datasets.

Our major contributions are as follows:

1. As far as we know, we have made the first attempt to explore a joint multiple ID and SF method with a global pointer, which can solve not only the nested and non-nested slots problem, but also the slot incoherence problem;

2. By constructing a multi-dimensional type-slot label interaction network, which can enhance the implicit association between intents and slots, to ensure the integrity of intent–slot information;

3. Our proposed new architecture achieves SOAT on multiple metrics on both public datasets, while having faster inference rates than other baseline models. While doing so, we carried out more in-depth ablation tests to examine the effects of various components on overall performance and serve as a guide for future model development.

## 2. Related Work

### 2.1. Intent Detection

ID is viewed as a classification task, which is a sub-field of text classification. Before the boom of deep learning, the first text classification method relied on hand-made feature spaces (such as one-hot representation) [28]. Nevertheless, with the continuous development of neural networks, various networks have been widely used in classification tasks. For instance, long short-term memory (LSTM) networks [29], CNN [30,31], GAT [26], BERT [32] and a robustly optimized BERT pre-training approach (Roberta) [33], etc. As the field of research develops, there are currently many techniques for pre-training word embedding, such as Glove [34] and Word2Vec [35], which are particularly trained on a large corpus to create unlabeled language and serve as a lexicon for various models. Kim et al. [36] employed enriched word vectors as the word embedding input of bi-directional LSTM for ID. Srivastava et al. [37] presented a hierarchical BERT architecture to detect the intention of utterances, which achieved good results.

### 2.2. Slot Filling

Sequence tagging is regarded as a method of SF decoding. CRF based on statistical methods has achieved great success in sequence labeling [38]. Deep learning has gained popularity recently. In particular, RNN and its variants show excellent performance on slot filling tasks and are superior to traditional machine learning methods. For instance, Wu et al. [39] proposed a slot-based language model with multi-modal interaction which obtained good results. Simminnet et al. [40] developed an architecture based on an attention mechanism and LSTM, where LSTM is applied to decoding slots, in light of the widespread use of attention mechanisms in NLP tasks. With the deepening of research, traditional machine learning approaches, such as CRF combined with neural networks,

may achieve better results. These statistical methods can be used as the decoding layer in various sequence labeling models. Saha et al. [41] introduced a model based on LSTM and CRF, which achieved a good performance on the ATIS (Airline Travel Information System) dataset.

### 2.3. Joint Model for Intent Detection and Slot Filling

In earlier studies, ID and SF were modeled as independent tasks. When these two highly related tasks are modeled separately, each module can not obtain enough information to complement itself, which ultimately leads to a poor effect on the whole process. Recently, many joint models have been presented to solve these problems. Joint models [3–21] were presented to consider the strong correlation between ID and SF with noteworthy success. Although these models apply to single-intent systems, they do not perform satisfactorily in multi-intent systems.

In a real-world dialogue system, the user's utterance usually contains multiple intentions and the corresponding semantic information. Joint models [22–25,42] were proposed to solve the multi-intent detection problem. Gangadharaiah and Naratanaswamy [22] proposed a multi-task framework for multi-intent detection and SF, which combines the loss functions of two modules without any inter-module information guidance between the two modules. Furthermore, graph neural networks (GNN) have also been employed for a variety of NLP tasks [24,25]. By constructing an interaction graph between tokens and different intentions, Qin et al. [24] presented an adaptive graph-interactive framework (AGIF) that models the strong correlation between slots and intents and achieves a further improved performance. However, they employed regression mode to decode the slots, resulting in a slower inference rate. In consideration of the inference speed, Qin et al. [25] explored a non-autoregressive framework that combines two local perception layers and a slot–intent interaction layer to build an intent–slot graph interaction network and achieved faster and better results on two multi-intent datasets. Although they implemented a cross-slot dependency modeling with an interaction graph, it still suffered from slot incoherence. Chen et al. [23] proposed a self-distilling architecture. The algorithm flow is as follows: firstly, pre-decode the slots; then, use the pre-decoded slot information to guide the decoding intent; next, use the decoded intent information to instruct the slot decoding; finally, use the decoded slot information as a soft label for the pre-decoded slots to form an optimization loop. An explicit slot–intent classifier was introduced to learn the many-to-one mapping between slots and intents to leverage the annotated data and capture the interaction between slots and intents, achieving a good performance [42]. However, the above two approaches cannot solve the problem of slot inconsistency.

The previous model ignored the slot nesting problem and treats them all as non-nested slots, which not only cannot tackle the problem of slot nesting but also has a poor performance and inference rate. Our approach is quite different from the previous methods. The proposed method not only solves the problem of slot incoherence caused by greedy decoding but also solves the problem of the slow inference rate of sequence annotation in regression mode, and is also able to solve the issue of slot nesting. It is the first work to discover, capture, and implement joint multi-intent detection and slot filling using global pointers.

## 3. Approach

In this section, we demonstrate the MTLN-GP model for the SLU task, as shown in Figure 2, which enables joint optimization of multi-intent detection and slot filling. We first define the span-based slot filling task. Then, we present the technical details of the proposed method.

### 3.1. Problem Definition

Extracting slots and intents from a given text is the goal of multiple ID and SF jobs. Intuitively, when given a sentence containing multiple intents and slots, the goal is to maximize the identification of intent and slot information. Let $U = \{u_{cls}, u_0, u_1, \cdots, u_{t-2}, u_{sep}\}$

be the possible spans in the sentence where $t - 2$ is the utterance length. The span $s$ is represented as $u[a : b]$, where $a$ and $b$ are the head and tail indexes, respectively. Meanwhile, we decoded the intent by multi-label classification. Finding every slot $s \in S$ and every intent $i \in I$ which correspond to the set of slots and the set of intents, respectively, is the aim of multiple ID and SF.

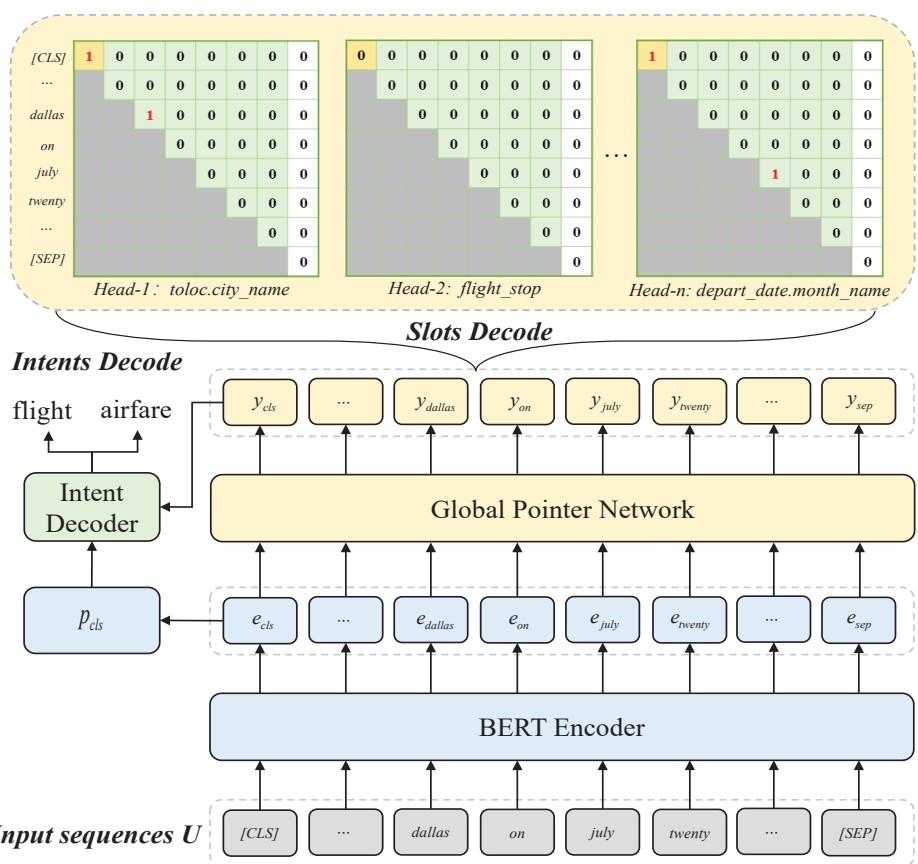

**Figure 2.** The architecture of MTLN-GP for joint multiple ID and SF consists mainly of BERT encoder, global pointer network, and intent decoder. The number "1" in red indicates that the slot tail pointer or slot type has been predicted.

### 3.2. BERT Encoder

To match BERT's input, we prepended [CLS] and appended [SEP]. Since the words of the text sequence may not exist in the dictionary of the pre-training model, the text sequence needed to be processed.

The workflow of the pre-training model was as follows: First of all, we utilized PieceTokenizer to divide the word of the text sequence into pieces, where $t$ is the utterance length. Next, for the input sequence $U$, it was input into the BERT pre-training model to capture the semantic components of the text to obtain $E \in \mathbb{R}^{t \times d}$ and $p_{cls} \in \mathbb{R}^{d}$, where $d$ and $p_{cls}$ are the hidden state dimension and the sentence-level semantic representation of the BERT output, respectively. Finally, we restored the sequence to the original sequence by simply applying a concatenation operation over piece representations.

$$U = \{u_{cls}, u_0, u_1, \cdots, u_{t-2}, u_{sep}\} \tag{1}$$

$$E = BERT(U) = \{e_{cls}, e_0, e_1, \cdots, e_{t-2}, e_{sep}\}. \tag{2}$$

### 3.3. Multi-Dimensional Type-Slot Label Interaction Network

As shown in Figure 3 , we built a multi-dimensional type-slot label interaction network $N \in \mathbb{R}^{n \times t \times t}$ to facilitate the implicit association of intents and slots, where $n$ and $t$

represent the total number of slot categories and the length represented by the BERT output, respectively. The dataset has $n$ different slot types, so different $n$ dimensions matrices need to be constructed. If the corresponding dimension has slots, [CLS] can be marked as "1", and the span tail with slots in the green area can be labeled as "1", otherwise it is marked as "0". The lower triangle of the matrix is marked in gray, indicating that the masking operation is performed in that part. Specifically, we decoded all the green fields of the multi-dimensional type-slot label interaction network for identifying the presence of slots and the gray fields for masking operations. The presence of marker "1" in the green area allows decoding of the slots with the coordinates of the marker to the main diagonal span, which can solve the slot incongruity problem.

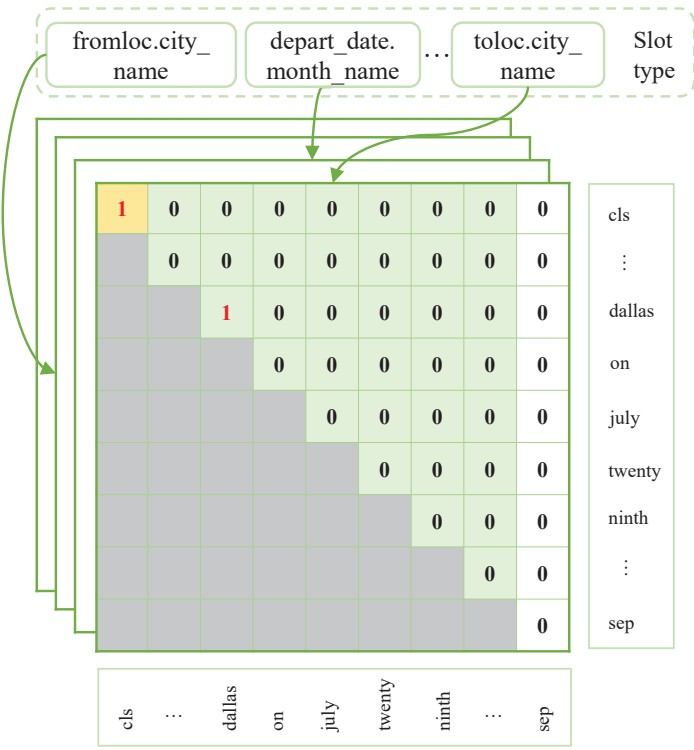

**Figure 3.** Illustrations of multi-dimensional type-slot label interaction network. Each element corresponds to a word pair. The gray, yellow, and green sections represent the lower triangle mask, type pointer, and slot tail pointer, respectively. The red number "1" indicates the type pointer or slot tail pointer at the corresponding position.

Specifically, the feature representation $E$ of the BERT output was fed into the global pointer network [27], and the decoding vectors $q_{i,\alpha}$ and $k_{i,\alpha}$ are shown in Equations (3) and (4). Since the traditional attention mechanism is not sensitive to the span slot and position information, it cannot capture the position information of the input sequence. Therefore, it was necessary to include rotational position embedding [43] in the decoding layer. The process of scoring the position of the $i$-th row and $j$-th column in each slot type matrix by adding a rotated position embedding, as shown in Equation (5), preserved the remote decay property and can be applied to linear attention, which satisfies $R_{j-i} = R_i R_j$.

$$q_{i,\alpha} = W_{q,\alpha} e_i + b_{q,\alpha} \tag{3}$$
$$k_{i,\alpha} = W_{k,\alpha} e_i + b_{k,\alpha} \tag{4}$$
$$s_\alpha(i,j) = (R_i q_{i,\alpha})^T (R_j k_{j,\alpha}) = q_{i,\alpha}^T R_{j-i} k_{j,\alpha}, \tag{5}$$

where $i$ and $\alpha$ denote the position of the $i$-th word in the sentence and the $\alpha$-th slot type in the slot category, respectively, where $t$ and $R$ are the length of the sentence and the relative

position embedding, respectively. $s_\alpha(i, j)$ represents the word attention score of the $i$-th row and the $j$-th column in the $\alpha$-th slot type matrix.

We fed the BERT output semantic representation $E = \{e_{cls}, e_0, e_1, \cdots, e_{t-2}, e_{sep}\}$ into the global pointer network [27] output $Y \in \mathbb{R}^{n \times t \times t}$, combined with our constructed multi-dimensional type-slot label interaction network and intention decoder bootstrap model to realize the implicit connection of intentions and slots, which can enhance the semantic association of intentions–slots, ensure the integrity of intention and slot information, and solve the slot nesting problem.

### 3.4. Intent Decoding

As shown in Figure 2, we took the probability value of the yellow position of each head, namely $O^I \in \mathbb{R}^n$. We employed $p_{cls}$ and $O^I$ to jointly decode the intent, which helped the implicit association of intent and slot to obtain fuller information about each other.

$$X = elu\left(W_y^I\left[O^I \mid\mid p_{cls}\right] + b_y^I\right) \tag{6}$$

$$Y^I = sigmoid\left(W_o^I\, X + b_o^I\right) \tag{7}$$

$$I = \left\{I_i \mid Y^I > p^I\right\}, \tag{8}$$

where $I_i$ represents the $i$-th intent in the set of intent labels. $W_y^I$, $W_o^I$, $b_y^I$ and $b_o^I$ are all trainable parameters for the intent decoder in this scenario. $\mid\mid$ indicates the aggregation operation, where *sigmoid* and *elu* are both activation functions. $p^I$ is the hyperparameters optimized by the validation set, and we set it to 0.55. Equation (8) represents the set of intentions with a probability greater than $p^I$.

### 3.5. Slot Decoding

As shown in Figure 2, we took all the probability values of the green area of each head, namely $O^S \in \mathbb{R}^{n \times (t-2) \times (t-2)}$. As shown in Equation (9), we compared it with the threshold $p^S$, and if there was a probability value more than the threshold $p^S$, we decoded all the slots by span.

$$S = \{s_{[a:b]} \mid O^S > p^S\}, \tag{9}$$

where $a$ and $b$ denote the indexes of the head and tail pointers after span decoding, respectively. $S$ represents the decoded the collection of slots, where $p^S$ is the hyperparameters optimized by the validation set, and we set it to 0.

### 3.6. Joint Optimization

As mentioned above, we built a multi-dimensional type-slot label interaction network that can employ slot filling as a binary classification task and whose loss function is a multi-label classification cross-entropy [27]:

$$\mathcal{L}_{slot} = \sum_{i,j=1}^{t} \sum_{\alpha=1}^{n} \left[\log\left(1 + \sum_{(i,j) \in P_\alpha} e^{-s_\alpha(i,j)}\right) + \log\left(1 + \sum_{(i,j) \in \Omega - P_\alpha} e^{s_\alpha(i,j)}\right)\right] \tag{10}$$

$$\Omega = \{(i,j) \mid 1 \leq i \leq j < t\}, \tag{11}$$

where $n$ and $t$ represent the total number of slot categories and the length represented by the BERT output, respectively. As shown in Figure 3, where $\Omega$ is the set of positions for all yellow and green regions under this sample, $\Omega$ contains $P$, which is the set of slot tail pointers and types pointers, where $i$ and $j$ represent the start and tail indexes of a span, and note that all $i$ must be less than or equal to $j$ in Equation (11).

This study describes multiple ID as a multi-label classification task. Equation (12) demonstrates that the loss function is a binary cross entropy.

$$\mathcal{L}_{intent} \triangleq - \sum_{i=1}^{t} \sum_{j=1}^{m} [y_i^{(j,I)} \log(\hat{y}_i^{(j,I)}) + (1 - y_i^{(j,I)}) \log(1 - \hat{y}_i^{(j,I)})], \tag{12}$$

where $y$ and $\hat{y}$ represent the golden intent label and the decoded sentence-level intent, respectively. $t$ and $m$ represent the length of the BERT output and the number of intent labels, respectively.

To jointly optimize multiple tasks, we combined the loss functions mentioned above, and the following is the final loss function for jointly optimizing multiple intent detection and slot filling tasks:

$$\mathcal{L}_{total\_loss} = \alpha \mathcal{L}_{slot} + \beta \mathcal{L}_{intent}, \tag{13}$$

where $\alpha$ and $\beta$ are the hyperparameters optimized by the validation set. $\alpha$ and $\beta$ are both set to 1.

## 4. Experimental and Analysis

### 4.1. Datasets

We evaluated the proposed approach with two multi-intent datasets—MixATIS and MixSNIPS. Table 1 demonstrates the statistical information of the two datasets.

**Table 1.** Statistics of two public datasets.

| Datasets | MixATIS | MixSNIPS |
|---|---|---|
| Training | 13,162 | 39,776 |
| Validation | 759 | 2198 |
| testing | 828 | 2199 |
| Intent | 18 | 7 |
| Slot | 78 | 39 |
| Vocabulary size | 827 | 9632 |

The MixATIS dataset [22,44] is a multi-intent dataset that includes audio recordings of individuals booking flights. It mainly contains the query of verbal information such as airline name, flight departure, airport code, etc. MixATIS includes 13,162 utterances for training, 756 for validation, and 828 for testing.

The MixSNIPS dataset [12,45] is a multi-intent dataset acquired from the Snips personal voice assistant. It mainly includes information such as querying the weather, playing movie music, movie reviews, searching for creative works and events, etc. MixSNIPS includes 39,776 utterances for training, 2198 for validation, and 2199 for testing.

### 4.2. Evaluation Metrics

We employed and extended an assessment measure [24] to assess our model. The F1-score and accuracy were applied for ID and SF tasks. Meanwhile, sentence accuracy refers to the percentage of utterances in the corpus whose slots and intentions are both correctly predicted [46], and it represents the total performance of the two tasks.

### 4.3. Experimental Settings

The experiment was conducted on a Linux server with NVIDIA GeForce RTX 3090, Python 3.7, and Tensorflow 2.4 framework [47]. All our models were based on BERT, which had 12 layers of transformer blocks, 768 hidden states and 12 self-attentive headers. The number of parameters of BERT-base is 110 million. The batch size of the MixATIS and MixSNIPS datasets is 24. The input dimension of the global pointer network is 128. The overall parameters of the model on the MixATIS and MixSNIPS datasets were 124 million and 117 million, respectively. The Adam optimizer with an initial learning rate of $5 \times 10^{-5}$ was applied to optimize the parameters. We selected the model that performed best on the validation set to evaluate the test set in all experiments.

To enhance the robustness of the model, the FGM [48] algorithm was used for adversarial training in the word embedding layer, which further enhanced the robustness of the model.

### 4.4. Baselines

The following are some typical baselines:

Attention BiRNN: A self-attention-based BiRNN joint model was presented by Liu and Lane [4], in which the intention is predicted by the weighted sum of hidden states.

Slot-Gated Atten: Goo et al. [5] presented a slot-gated mechanism that directly takes into account the relationship between SF and ID.

Bi-Model: Wang et al. [6] introduced a bidirectional model, which uses BiRNN to decode intent and slot tasks, respectively, and share the hidden state information of each time step between two decoders.

SF-ID Network: Niu et al. [7] presented an SF-ID architecture, which provides a direct connection between intent and slot, allowing them to promote one another.

Stack-Propagation: Qin et al. [8] introduced a stack-propagation architecture that guides the SF task by combining the decoding intent with the encoding information.

Joint Multiple ID-SF: To assist slot decoding, Gangadharaiah and Narayanaswamy [22] developed a slot-gated model with attention that incorporates slot context vector and intent context vector as slot gating.

AGIF: Qin et al. [24] created an adaptive intent–slot graph interaction network based on GNN for SF that uses decoded intent and token sequence as nodes.

GL-GIN: To address the issue of slot inconsistency and realize the interaction between intents and slots, Qin et al. [25] suggested a fast and accurate non-autoregressive model based on GAT that contains a global–local graph interaction network. The model has looked excellent, with the rate of inference increasing.

Joint BERT: Chen et al. [9] developed a BERT-based joint model that uses *CLS* to decode the intent and the token sequence to decode the slot directly. This model achieves a good performance on multiple indicators.

SDJN+BERT: Chen et al. [23] proposed a self-distillation architecture that gives intent and slot information to each other to achieve cyclic optimization and implements self-distillation by treating the decoded slots as soft labels for the pre-decoded slots.

### 4.5. Main Results

We reproduced a baseline model on two multi-intent datasets to examine the effectiveness of our proposed approach. For the *JointBERT* [9] model, the original author did not release their code, so we reproduced it. In the *JointBERT* model, we employed binary cross-entropy loss and changed the intent decoder, replacing softmax as sigmoid.

The results of 10 times validation on the baseline model and our model were subjected to statistical significance testing using the Student's t-test with a significance level of 0.05. Our novel method achieves SOTA performance on multiple metrics on MixATIS and MixSNIPS datasets, as shown in Table 2. The current best results on the MixATIS dataset were 88.3%, 78.0%, and 46.3% for slot F1, intent Acc, and sentence Acc, respectively. On the MixSNIPS dataset, the best results for slot F1, intention Acc, and sentence Acc were 95.6%, 96.7%, and 79.8%, respectively. From the results, we observe that: (1) For slot F1, the proposed model is 0.1% and 1.1% higher than the current optimal baseline GL-GIN [25] and Joint BERT [9] model, respectively. This is because the global pointer model can solve the problem of slot incoherence more effectively. (2) For intent Acc, the proposed model is 1.6% and 1.2% higher than the current optimal baseline SDJN+BERT [23] model. This is due to the fact that the constructed multi-dimensional type-slot label interaction network can provide more rich semantic information for intent decoding. (3) For sentence accuracy, the proposed model is significantly higher than the current optimal results by 3.1% and 4.5%. This is because the constructed multi-dimensional type-slot label interaction network enhances the implicit association of intentions and slots, allowing sufficient information on each other to improve the overall performance of the model.

**Table 2.** Results of joint training on the MixATIS and MixSNIPS datasets. The values with * and bolded indicate that our framework has improved on all of its baselines (%).

| Model | MixATIS | | | MixSNIPS | | |
|---|---|---|---|---|---|---|
| | Slot(F1) | Intent(Acc) | Overall(Acc) | Slot(F1) | Intent(Acc) | Overall(Acc) |
| Attention BiRNN [4] | 86.4 | 74.6 | 39.1 | 89.4 | 95.4 | 59.5 |
| Slot-Gated [5] | 87.7 | 63.9 | 35.3 | 87.9 | 94.6 | 55.4 |
| Bi-Model [6] | 83.9 | 70.3 | 34.4 | 90.7 | 95.6 | 63.4 |
| SF-ID [7] | 87.4 | 66.2 | 34.9 | 90.6 | 95.0 | 59.9 |
| Stack-ropagation [8] | 87.8 | 72.1 | 40.1 | 94.2 | 96.0 | 72.9 |
| Joint Multiple ID-SF [22] | 84.6 | 73.4 | 36.1 | 90.6 | 95.1 | 62.9 |
| AGIF [24] | 86.7 | 74.4 | 40.8 | 94.2 | 95.1 | 74.2 |
| GL-GIN [25] | 88.3 | 76.3 | 43.5 | 94.9 | 95.6 | 75.4 |
| SDJN [23] | 88.2 | 77.1 | 44.6 | 94.4 | 96.5 | 75.7 |
| SDJN+BERT [23] | 87.5 | 78.0 | 46.3 | 95.4 | 96.7 | 79.3 |
| Joint BERT [9] | 86.1 | 74.8 | 44.8 | 95.6 | 96.2 | 79.8 |
| MTLN-GP (ours) | **88.4 *** | **79.6 *** | **49.4 *** | **96.7 *** | **97.9 *** | **84.3 *** |

It can be seen that using the global pointer approach to decode the tail pointer of slots can decode all slots, which can solve the issues of slot inconsistency in the decoding of the BIO labeling approach and improve the performance of slot filling. The multi-dimensional type-slot label interaction network can realize slot and slot type interaction, provide slot information for downstream intent decoding, promote the implicit association of intent and slot, and greatly improve the intent accuracy rate. Meanwhile, the accuracy of the whole sentence is also improved significantly, which is attributed to the constructed multi-dimensional type-slot label interaction network.

We attribute the above benefits to our approach to constructing the type-slot label interaction network, which implicitly helps the model capture more explicit and high-confidence correspondence between intent and slots to reduce the propagation of errors and bring about significant improvements.

### 4.6. Analysis

#### 4.6.1. Speedup

We followed the same setup as in [24] to compute the inference speedups. As shown in Table 3, the MTLN-GP model has a significant increase in inference rate compared to several typical baseline models. Compared with stack-propagation [8], joint multiple ID-SF [22], and AGIF [24], the global pointer can avoid the inference delay caused by the word-by-word decoding of the regression mode. Meanwhile, compared with GL-GIN [25], the proposed method does not need to spend considerable time on building graph networks and softmax computation.

The combination of Tables 2 and 3 shows that our model not only has a significant speedup but also a good performance. Our model is valuable due to its time complexity and performance.

**Table 3.** Speed comparison.

| Model | Decode Latency(s) | Speedup |
|---|---|---|
| Stack-propagation [8] | 34.5 | 1.4× |
| Joint Multiple ID-SF [22] | 45.3 | 1.1× |
| AGIF [24] | 48.5 | 1.0× |
| GL-GIN [25] | 4.2 | 11.5× |
| Joint BERT [9] | 2.3 | 21.1× |
| MTLN-GP | 3.1 | 15.6× |

### 4.6.2. Ablation Experiments

The MTLN-GP model has improved a number of metrics on two benchmark datasets, according to experimental findings. However, we need to know why the situation has improved. In this section, we provide a detailed ablation analysis to investigate the contribution of the sub-net to our model.

With the ablation that follows, we investigated the efficacy of MTLN. As shown in Figure 3, we set the "1" in the yellow area under each slot type dimension to "0" and left the others unchanged, namely MSLN-GP. Table 4 demonstrates that our technique is enhanced in multiple metrics. This is because the yellow area of the proposed multidimensional type-slot label interaction network integrates the information of each type of slot and can facilitate intent decoding.

**Table 4.** On two multi-intent datasets, we conducted an ablation comparison of our proposed approach (%).

| Model | MixATIS | | | MixSNIPS | | |
|---|---|---|---|---|---|---|
| | Slot(F1) | Intent(Acc) | Overall(Acc) | Slot(F1) | Intent(Acc) | Overall(Acc) |
| Joint BERT [9] | 86.1 | 74.8 | 44.8 | 95.6 | 96.2 | 79.8 |
| MSLN-GP | 88.2 | 78.3 | 47.2 | 95.9 | 97.1 | 82.6 |
| MTLN-GP *w/o Joint* | 87.9 | 78.4 | 47.8 | 96.3 | 97.4 | 83.3 |
| MTLN-GP by Max Pooling | 88.2 | 79.3 | 48.9 | 96.7 | 97.6 | 84.1 |
| MTLN-GP by Mean Pooling | 88.1 | 79.4 | 49.1 | 96.6 | 97.6 | 83.9 |
| MTLN-GP | **88.4** | **79.6** | **49.4** | **96.7** | **97.9** | **84.3** |

Bolded indicates the best performance of the model.

Next, we directly employed the $p_{cls}$ output from BERT for decoding the intent instead of combining the $p_{cls}$ with the global pointer network output information for decoding the intent, namely MTLN-GP *w/o Joint*. As shown in Table 4, the performance of the MTLN-GP model increases in all metrics compared to the MTLN-GP *w/o Joint* model, which is due to the ability of the multidimensional type-slot label interaction network to facilitate intent decoding and contribute to the implicit association between intent and slot, enabling each other to obtain complementary information.

Finally, we evaluated the impact of the feature fusion method on the performance of the global pointer output $Y$. The following two experiments were conducted for the acquisition of $O^I$ in Equation (6): first, the 2-D Max pooling was applied to $Y$, namely MTLN-GP by Max Pooling; second, the 2-D AVG pooling was applied to $Y$, namely MTLN-GP by Mean Pooling. As can be seen from Table 4, our method outperforms both approaches. This is because the [CLS] in the multi-dimensional type-slot label interaction network we designed contains information processed for each slot type, while the pooling operation introduces some noisy information.

### 4.6.3. Single-Intent and Multi-Intent Analysis

In addition, we wanted to know the effect of single-intent and multi-intent samples on the proposed model. The test set was divided into single-intent and multi-intent based on the number of intentions in the utterances, and the experimental results are shown in Figure 4. The samples of the MixATIS test set were 143 and 685 for single-intent and multi-intent, respectively, with 450 and 1749 for the MixSNIPS test set, respectively. As can be seen from the results: (1) The single-intent and multi-intent samples are essentially equal in slot F1, which indicates that the proposed model addresses the slot incoherence problem well with global pointers; (2) In terms of intent accuracy, MixATIS has a higher gap relative to MixSNIPS in both samples, which may be caused by the unbalanced distribution of intent labels in MixATIS, with only 17 intents in the training set and 18 intents in the test set. Generally, the proposed model can nicely treat both single-intent and multi-intent samples; (3) For sentence accuracy, single-intent samples work better than multi-intent samples. Nevertheless, it is a predictable scenario because the single-intent task is relatively easier

than the multi-intent task. In summary, the proposed method obtains a good performance in both samples.

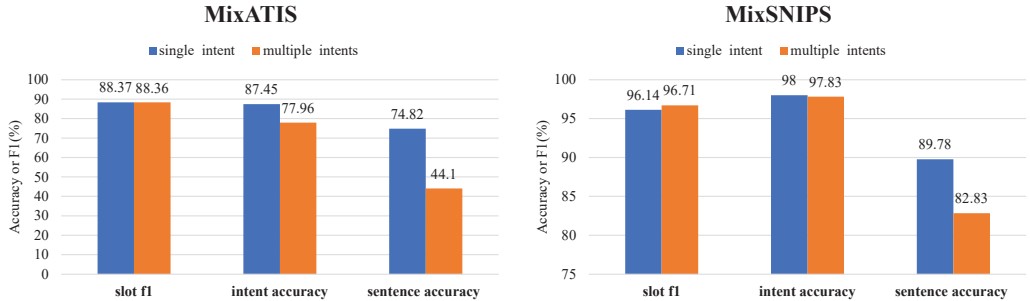

**Figure 4.** Accuracy of single-intent and multi-intent comparison plots.

### 4.6.4. Error Analysis

The experimental findings demonstrate the MTLN-GP model's high performance in terms of inference rate and performance, reaching SOAT for a number of metrics on two benchmark datasets. Nevertheless, the results are still relatively poor on some datasets. We analyzed the reasons in the following respects:

1.  The data set is limited and the distribution of categories is unbalanced. Some slot types are not present in the training set, but appear frequently in the test set. Meanwhile, out of vocabulary (OOV) appears in the test set;
2.  The second reason is the annotation error. As shown in Table 5, the intent of the first sample was manually marked as "aircraft". Nevertheless, we note that the question refers to the number of different aircraft types. By observation, the red word in the second sample should be labeled "*toloc.city_name*", but it was marked as "*city_name*". Meanwhile, the results predicted by the MTLN-GP model are generally consistent with the actual observations;
3.  Irrelevant information can mislead the prediction results. As shown in Table 5, the proposed model is misled by the slot information and therefore outputs the wrong intention. The actual content of the sentence is "what days of the week".

**Table 5.** The golden label is also included in this example of three testing samples generated by our method on the MixATIS dataset. ⋆ Indicates that no slot exists at this position.

| Model | Golden Slots | Golden Intent |
|---|---|---|
| Text | at the charlotte airport how many different types of aircraft are there for us air | |
| Golden | ⋆ ⋆ city_name city_name ⋆ ⋆ ⋆ ⋆ ⋆ ⋆ ⋆ ⋆ airline_name airline_name | atis_aircraft |
| MTLN-GP | ⋆ ⋆ city_name city_name ⋆ ⋆ ⋆ ⋆ ⋆ ⋆ ⋆ ⋆ airline_name airline_name | atis_quantity |
| Text | list the distance in miles from boston airport to downtown boston | |
| Golden | ⋆ ⋆ ⋆ ⋆ ⋆ ⋆ fromloc.airport_name fromloc.airport_name ⋆ ⋆ city_name | atis_distance |
| MTLN-GP | ⋆ ⋆ ⋆ ⋆ ⋆ ⋆ fromloc.airport_name fromloc.airport_name ⋆ ⋆ toloc.city_name | atis_distance |
| Text | what days of the week do flights from san jose to nashville fly on | |
| Golden | ⋆ ⋆ ⋆ ⋆ ⋆ ⋆ ⋆ ⋆ fromloc.city_name fromloc.city_name ⋆ toloc.city_name ⋆ ⋆ | atis_day_name |
| MTLN-GP | ⋆ ⋆ ⋆ ⋆ ⋆ ⋆ ⋆ ⋆ fromloc.city_name fromloc.city_name ⋆ toloc.city_name ⋆ ⋆ | atis_flight |

Red font indicates data labeling errors or prediction errors.

## 5. Conclusions

In this article, a joint model for solving nested and non-nested slots based on global pointers was proposed to address the problem that the existing models do not have high enough performance and inference rates. By considering the implicit correlation between intents and slots, a novel multi-dimensional type-slot labeling interaction network was designed. At the same time, the introduction of the global pointer network can not only

address the problem of inconsistent slots, but also handle both nested and non-nested slots, which lays a certain foundation for solving nested slots subsequently. We are the first to propose the use of pointer networks for this task, and its performance and inference rate are guaranteed. Experiments on two public multi-intent datasets validate the effectiveness of the algorithm.

As a further step, we will continue to study how to improve the accuracy of the model and apply it to practical engineering.

**Author Contributions:** Conceptualization, X.W. and W.Z.; methodology, X.W.; software, X.W.; validation, X.W. and Y.W.; formal analysis, X.W. and S.F.; investigation, X.W. and S.F.; resources, W.Z. and S.F.; data curation, X.W.; writing—original draft preparation, X.W.; writing—review and editing, W.Z., X.W. and M.H.; supervision, W.Z. and M.H.; project administration, W.Z. and M.H.; funding acquisition, W.Z. and M.H. All authors have read and agreed to the published version of the manuscript.

**Funding:** This work was supported in part by the National Natural Science Foundation of China (82260362, 62241202), in part by the National Key R&D Program of China (2021ZD0111000).

**Institutional Review Board Statement:** Not applicable.

**Informed Consent Statement:** Not applicable.

**Data Availability Statement:** Not applicable.

**Conflicts of Interest:** The authors declare no conflict of interest.

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
