# Peer review of "A Unified Approach to Nested and Non-Nested Slots for Spoken Language Understanding"

_electronics, doi:10.3390/electronics12071748_

Round 1

Reviewer 1 Report

This paper proposed a unified model for spoken language understanding. The main idea is to leverage the nested and non-nested slot and multi-intent joint modeling. The idea is interesting. I only have several concerns.

1) The formulation and equations are not well organized. Some symbols are not defined. Please check all the symbols and equations to make it clear.

2) It is better to show more qualitative results to help better evaluation of the proposed method.

3) Missing related slot-based language models. For example, "Switchable Novel Object Captioner, TPAMI 23" proposed a slot-based language model with multi-modal interaction, which shares the same idea of the proposed method. The authors should clearly discuss the difference from this paper in the related work section.

4) What does "Speedup" in Table 3 mean? How is it calculated? Is it the smaller the better?

Author Response

Dear Editors and Reviewers,

Thank you for your letter and for the reviewers’ comments concerning our manuscript entitled “A Unified Approach to Nested and Non-nested Slots for Spoken Language Understanding” (Manuscript ID: electronics-2263813). Those comments are all valuable and very helpful for revising and improving our paper, as well as the important guiding significance to our research. We have studied the comments carefully and have made a correction which we hope meets with approval. Revised portions are marked in red in this manuscript. The main corrections in the paper and the responses to the reviewer’s comments are as follows:

Reviewer 2 Report

The authors of the work presented nested and non-nested slots for spoken language understanding. I have some concerns/comments on this work, as bellow.

1. section 2 named relation work should change to related work.

2. References 26,27 are not in the right format. for example, there is no date on the reference, as well as references 32,33, .. check all the references

3.  Explain more about the resian for putting a global pointer network in the model and how it works.

4. explain more about why MTLN-GP has higher performance than  others,

5. In most of the work the authors compared with, there are not well-defined references without details how can we relay on those papers ? (22,23,24,9)

6. I strongly recommend the work compared with some well know work in well-known journals/conferences.

Author Response

(The authors gave the same response as above.)

Round 2

Reviewer 2 Report

Thank you for addressing all the review comments and for the revised paper. Well done.